# Influence of Exposure Conditions and Particulate Deposition on Anodized Aluminum Corrosion

Isabel Rute Fontinha *  and Elsa Eustáquio

Laboratório Nacional de Engenharia Civil, Avenida do Brasil 101, 1700-066 Lisboa, Portugal
* Correspondence: rfontinha@lnec.pt

**Abstract:** Anodizing is commonly used for corrosion protection of aluminum and its alloys in the construction industry. The anodic aluminum oxide (AAO) coating has a high ability to prevent the development of extensive pitting corrosion in aluminum substrates, particularly in marine sites, as was observed during a 10-year atmospheric corrosion study carried out in several marine and industrial sites. However, this study also evidenced that this coating can be highly affected by the deposition of particulate material in industrial polluted environments, sometimes in unexpected ways. This study presents information on the atmospheric corrosion of anodized aluminum exposed at two different chemical industrial complexes: a fertilizer production plant and a pulp and paper mill. Visual assessment of surface changes, pitting depth and mass variation with exposure were determined to quantify the degradation suffered. Additionally, SEM/EDS analyses were carried out on the exposed surfaces. Based on the results obtained, the role played by the deposition of airborne particles present in the two environments with respect to the type and level of damage observed is discussed. Deposits of roasted pyrite ash and phosphates or of wood chips and lime particles enhanced pitting corrosion or caused dissolution of the AAO coating.

**Keywords:** atmospheric corrosion tests; anodized aluminum; anodic aluminum oxide (AAO); pitting; particle pollutants; industrial environment; fertilizer plant; pulp and paper industry



## 1. Introduction

Aluminum and its alloys are widely used in several applications due to their interesting specific mechanical properties and corrosion resistance. A great number of these applications occur in the construction industry, for instance, in building framework profiles and cladding sheets, in which aluminum components are exposed to the atmosphere [1]. Although aluminum alloys commonly used for architectural applications (usually EN AW 6060, 6063 or 5005 alloys) present good corrosion resistance, due to the spontaneous oxide layer that forms on their surfaces, they are prone to developing corrosion in the form of pitting and other kinds of localized corrosion, which can be extensive in marine and industrial atmospheres [2–10]. Therefore, not only to mitigate corrosion processes, but also because maintaining a good aesthetic appearance is of prime importance, architectural aluminum-based building elements are commonly used with anticorrosive coatings, such as paints (powder coatings) and anodic oxide coatings.

Architectural anodization is an electrochemical process in which an aluminum alloy is anodically polarized in a diluted sulfuric acid-based solution (at temperatures close to 20 °C) to produce a protective oxide layer much thicker than the natural oxide film that forms on these types of alloys. Atmospheric corrosion studies carried out in a wide variety of atmospheres have confirmed the superior ability of these types of anodic oxide coatings to protect aluminum against corrosion in aggressive environments, such as marine and heavily polluted atmospheres, provided that a certain critical thickness is exceeded and an appropriate fabrication procedure is used [2–4,11]. According to these studies and some surveys performed in buildings, in urban atmospheres that are usually of moderate

corrosivity, a durability of about 100 years [12–14] can be predicted for anodic coatings, considering the thicknesses usually applied in architecture. The significant role played by anodic coating thickness in the corrosion protection of aluminum in marine or industrial atmospheres has been evidenced by several field studies. Therefore, anodic coating thickness classes, to be used in accordance with the atmospheric conditions of the exposure site, are often stipulated in national standards and technical documents [15–17].

Usually, the corrosion performance of aluminum is related to the time of wetness and atmospheric levels of chloride species and $SO_x$-type gaseous pollutants—parameters used to attribute corrosivity classes to an exposure site [18]. Aluminum corrosion (mainly in a localized form, such as pitting) increases with the chloride level and humidity of the atmosphere. $SO_x$ pollution is more significant when chloride salinity is also present (as in marine–industrial atmospheres) [2,3,9,10,14,19], although its presence in other atmospheres can induce aesthetic degradations (stains, darkening, etc.) and even generalized corrosion [14,20]. The corrosion resistance of anodized aluminum has also been related to chloride salinity and $SO_x$ gaseous pollutants in the atmosphere, although in a slightly different manner, since both can damage the anodic coatings. Anodized aluminum is much less affected by atmospheric chloride salinity than aluminum, but the detrimental effects of those compounds are highly enhanced by $SO_x$ gaseous pollutants in wet environments [3,4].

In general, it has been observed that only anodic coatings with a thickness above 25 μm can last ten years without showing pitting corrosion in the most aggressive industrial atmospheres. Anodic coatings of lower thicknesses will develop corrosion earlier and can develop pitting corrosion in pure marine atmospheres with high salinity levels, but these processes will require much longer exposure times to occur than in the case of bare aluminum. Increase in the thickness of the anodic coating will delay the initiation of the corrosion process and its extension [2–5,11].

Other factors that were found to affect anodized aluminum corrosion performance in natural environments were: coating sealing quality (sulfuric acid anodization produces porous coatings) and the presence of physical defects, such as cracking; iron content and the consequent number and size of intermetallic particles in the alloy; and the degree of soiling and the nature of the deposited particles [2,4,11,19,21]. The latter parameter was pointed out as the reason for the unusual behavior shown by anodized aluminum specimens exposed at Lima station, which was unexpected based on the corrosivity (assessed by chloride and $SO_2$ deposition rates). In that site, there was a heavy deposition of carbonaceous particles (soot), which are highly electropositive and hence able to activate local corrosion cells on the surface of highly electronegative aluminum [2].

The effect of the accumulation of solid deposits on anodized aluminum surfaces has been addressed in very few studies [2,4,22]. The presence of particles on such surfaces can be highly detrimental to anodic coating performance and can lead to their premature corrosion if they promote the formation of acid (pH < 4) or alkaline (pH > 8.5) condensates, in which alumina cannot remain passive [3].

The present work describes two case studies in which exposure to airborne particulate matter in industrial environments contributed to a higher level of degradation of anodized aluminum than would be expected considering only atmospheric pollution. In one case, these deposits, of an acidic nature, enhanced pitting corrosion, while in the other case the deposits were of an alkaline nature and caused extensive dissolution of the anodic coating. The main constituents of these deposits were related to the effluents and waste from the industrial chemical processes occurring nearby in sulfuric acid and fertilizer plants and a pulp and paper mill.

## 2. Materials and Methods

The materials used in this study belong to an extensive natural exposure testing program that has been carried out by the Portuguese National Laboratory for Civil Engineering (LNEC) in several marine, industrial and urban environments, comprising different metallic materials commonly used in building components: aluminum and anodized aluminum,

zinc and several types of zinc-coated steel, as well as copper. The program started in 1985, and the majority of the data were gathered over ten years of exposure [23,24]. After that period, some tests sites had to be shut down, namely, the most corrosive ones. However, in the less corrosive test sites, the most corrosion-resistant materials, including anodized aluminum, remained exposed.

The data presented here are relative to materials exposed for ten years at two of the above-mentioned study's test sites that have been dismantled and were located inside industrial parks. The analyses carried out focused on evaluating the influence of the exposure conditions and particulate matter deposition on the corrosion/degradation processes that occurred in the anodized aluminum specimens.

### 2.1. Materials

Aluminum flat sheets of 1050 alloy with a minimum purity of 99.5% and the following maximum percentage concentrations for the alloying elements: 0.17 Si, 0.15 Fe, 0.01 Cu, 0.18 Ti and 0.005 Mn, were clear-anodized and cut to produce 12 cm × 20 cm × 0.1 cm test specimens. Each one was marked by drilling holes (diameter: 2 mm) at different positions for identification. Specimens of the same aluminum sheet without anodization were also prepared. The anodization process was carried out in industrial conditions, using the sulfuric acid process, following Qualanod quality-label specifications for architectural coatings, which required proper sealing of the anodic coating porous layer [17]. Different anodization times were applied to produce anodic oxidation coatings of three thickness ranges: 15 μm–18 μm, 20 μm–25 μm and 30 μm–35 μm, covering the usual range of thicknesses that can be found in architectural exterior applications.

### 2.2. Exposure Conditions

The aluminum test specimens in the bare condition and anodized with the three above-mentioned coating thickness ranges were mounted on porcelain insulators, on painted galvanized steel racks, at an angle of 45° (Figure 1), facing south, according to ISO 8565 [25], at two industrial test sites. The total duration of the exposure was 10 years. Test specimens (triplicates for gravimetric measurements and singles for surface analysis) were collected after 6 months, 1 year, 3 years, 5 years and 10 years of exposure.

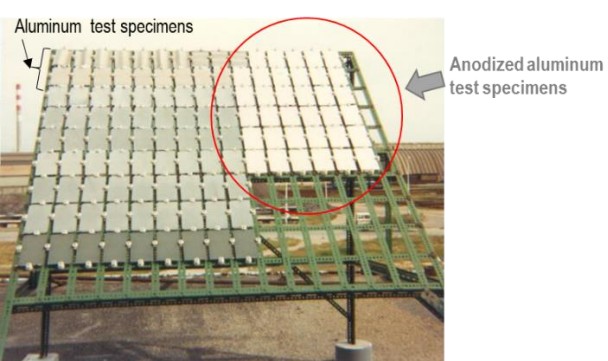

**Figure 1.** General view of one of the exposure racks (e.g., at the Barreiro test site) after installation.

The test sites were located in two of the most polluted industrial sites in Portugal at the time [26] (Figure 2):

- Barreiro: a conglomerate of chemical industries, including sulfuric acid and fertilizers production plants, near the sea at a river estuary in a highly populated area (Lisbon); and
- Rodão: a pulp and paper mill complex, inland, in a very sparsely populated area.

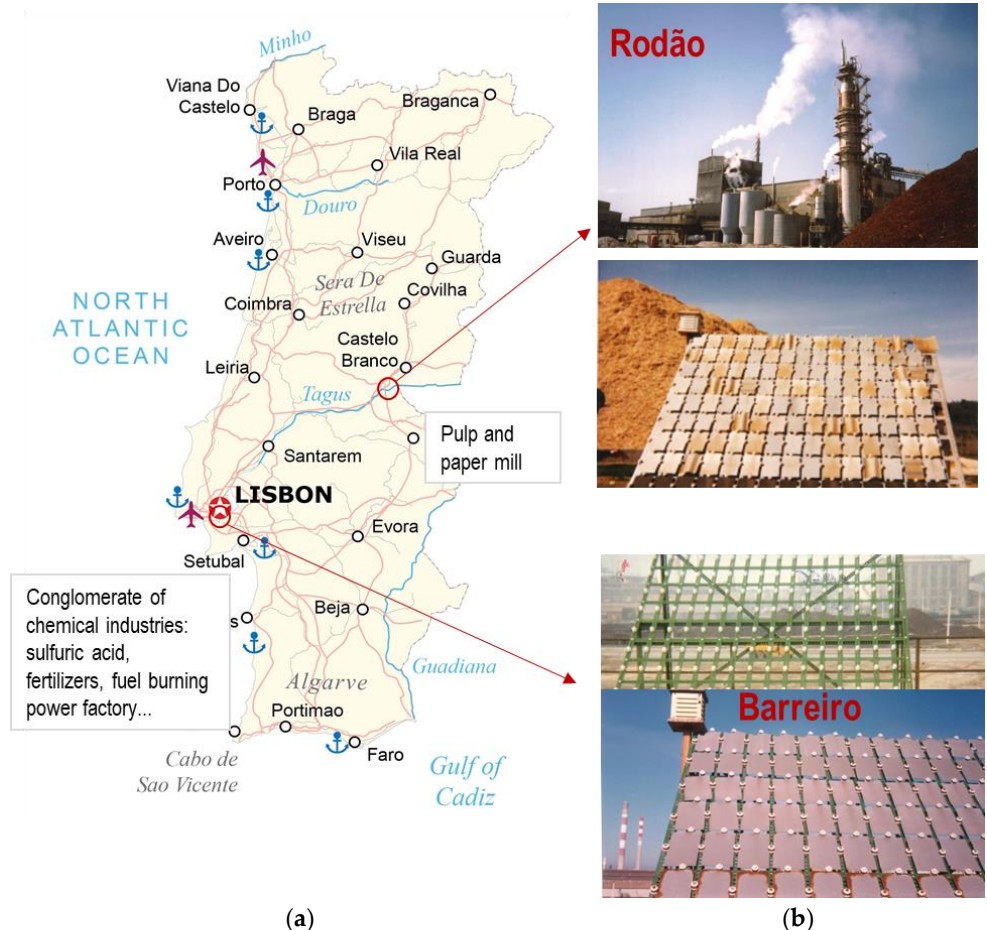

**Figure 2.** Exposure test sites. (**a**) Map of Portugal [27] with the locations of the test sites (**b**) Photos of the exposure sites: Rodão (top) and Barreiro (bottom), which show solid waste deposits of industrial waste near the racks and the coloration of the test specimens' surfaces due to the deposition and accumulation of waste particles.

At this last site, the exposure racks were initially placed very close to wood-chip deposits, the deposition of which was clearly visible on the test specimens' surfaces, as shown in Figure 2. The racks were relocated after 1 year of exposure.

Air samples for the analysis of $SO_2$ and chloride contents were collected continuously, at both sites, during 7 to 8 years of exposure, according to the methodology described in ISO 9225 [18]. $SO_2$ was determined as sulfate by the lead dioxide adsorption method. Chlorides were collected via the wet candle method and evaluated by potentiometric titration. Meteorological data relative to the exposure period were also collected for both sites from weather monitoring stations nearby and later analyzed. Times of wetness (TOW) were estimated based on relative humidity (HR) and temperature (T) data available from the Meteorological Institute library (number of hours with HR > 80% and T > 0 °C).

Table 1 shows the pollution levels and climatological characteristics of the testing sites where the exposures were carried out. The respective corrosivity categories estimated from environmental and corrosion data (the latter obtained from the exposure of the bare aluminum specimens), according to ISO standards [28–30], are also shown.

**Table 1.** Pollution levels, climatological characteristics and corrosivity categories of the exposure sites based on environmental data and on aluminum corrosion rates [23].

| Exposure Site | Temperature (Yearly Av.) (°C) | TOW (h·y$^{-1}$/Annual Fraction) | SO$_2$ (mg·m$^{-2}$·d$^{-1}$) | Chlorides (mg·m$^{-2}$·d$^{-1}$) | Corrosivity Category | | | |
|---|---|---|---|---|---|---|---|---|
| | | | | | Environ. (ISO 9223:1991) | Al Corrosion Rate (1 y/ISO 9223:2012) (10 y/ISO 9224:2012) | | |
| Barreiro | 14.8 | 3388/0.39 | 136 | 38 | C4/C5 | 1 y | 20.3 g·m$^{-2}$ | CX |
| | | | | | | 10 y | 71.0 g·m$^{-2}$ | >C5 |
| Rodão | 15.1 | 1871/0.22 | 21 | 5 | C3 | 1 y | 1.2 g·m$^{-2}$ | C3 |
| | | | | | | 10 y | 13.2 g·m$^{-2}$ | C4 |

Barreiro has a highly polluted industrial atmosphere due to the high average SO$_2$ deposition rates, which, during the first five years of exposure, were quite high (200 mg·m$^{-2}$·d$^{-1}$) and can thus account for the elevated corrosion rates exhibited by the bare aluminum specimens. However, SO$_2$ deposition rates decreased significantly in the following years (to 50 mg·m$^{-2}$·d$^{-1}$) due to several factories closing.

The SO$_2$ deposition rates measured at the Rodão test site were much lower than at Barreiro and, consequently, the aluminum corrosion rates were also. SO$_2$ pollution levels at Rodão show that this test site has the characteristics of an urban environment with moderate pollution, although it is located in a rural area.

### 2.3. Corrosion Performance Assessment Methodology

Visual assessment of surface changes, pitting depth and mass variation with exposure were determined to quantify the degradation suffered.

Optical and electronic SEM microscopy observations complemented with EDS analyses were carried out on the test specimens to characterize the attack morphology and further understand the degradation process suffered.

A Stereoscopic loupe Olympus SZH (OM) and a metallographic microscope Olympus PMG3 were used for optical observation (OM) of the specimens' surfaces and cross sections and for the measurement of pit depths, the latter carried out using a procedure based on ISO 1463 [31]. Scanning electron micrographs and SEM/EDS data were obtained with a scanning electronic microscope (SEM) Tescan Mira3 coupled with a Bruker energy dispersive X-ray spectrometer (EDS) XFlash 6|30, using an electron beam voltage of 20 kV. Some test samples had to be sputtered with a thin gold layer to allow for good observations via SEM.

Gravimetric determinations: before exposure, all test specimens were washed in distilled water, solvent-degreased, dried and weighed. The test specimens used for gravimetric determinations were weighed after exposure, cleaned and re-weighed. The following cleaning procedures, based on ISO 8407 [32], were used:

- Bare aluminum: after exposure, loose corrosion products and deposits were removed from the specimens by washing with water and a neutral soap; the specimens were then lightly brushed, washed again with distilled water and immersed in a phosphochromic solution of 2% Cr$_2$O$_3$ and 35 mL of phosphoric acid ($\rho$ = 1.7 g/mL) at 80 °C to 85 °C for 5 min.
- Anodized aluminum:

1. *Step I*: after exposure, loose corrosion products and deposits were removed by washing the specimens in water with a neutral soap; the specimens were lightly brushed, washed again with distilled water and dried, then weighed;
2. *Step II*: immersion, for 2.5 min, of the previously cleaned (*Step I*) specimens in 65% nitric acid (HNO$_3$, $\rho \geq 1.39$ g/mL), at room temperature; the specimens were then washed with distilled water and dried, then weighed;
3. If necessary, *Step II* was repeated. Number of *Step II* cleaning cycles: 1–3.

The *Step II* cleaning procedure was introduced because the *Step I* cleaning procedure did not remove most of the dirt present on the anodized aluminum specimen surfaces. It

was established based on ISO 8407 guidelines and assessed using unexposed anodized specimens that had been kept stored in the laboratory. This cleaning procedure, which has a minor impact on anodic coatings in good condition, revealed more realistically the attack suffered by the anodic layers due to atmospheric exposure than sole use of the more conservative procedure (*Step I*), as has been used in other studies [2].

The assessment of the damage inflicted by atmospheric exposure to the anodic coatings was complemented by carrying out measurements of the admittances of the anodic layers, according to ISO 2931 [33]. This "admittance test" is commonly used as a quality control test for assessing the sealing degrees of architectural aluminum anodic coatings [34]. The admittance (a reciprocal of the impedance) at a frequency of 1000 Hz can be related to the corrosion resistance of the coating and has been used as a "measure" of coating integrity [35].

## 3. Results

### 3.1. Surface Aspect Modifications and Corrosion Processes

Figure 3 shows the typical visual aspects of the skyward (front) and downward (back) faces of the anodized aluminum specimens from both test sites after ten years' exposure.

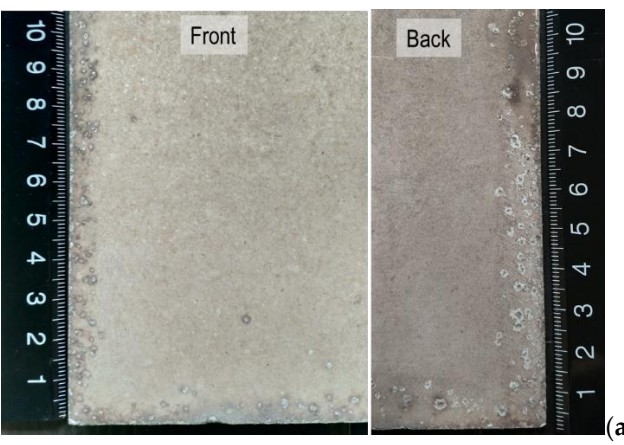
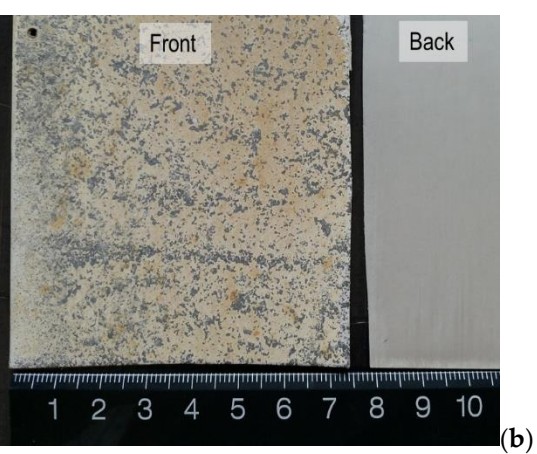

**Figure 3.** Details of the visual aspects of the front (exposed) and back surfaces after ten years of exposure at the test sites: (**a**) Barreiro (industrial–marine); (**b**) Rodão (industrial).

After ten years of exposure in the industrial–marine test site at Barreiro, the anodized aluminum specimen surfaces (both faces) acquired a reddish-grey hue (this was observed for all the exposed materials, as can be seen in Figure 2) and showed deep pits mostly on the edges and back faces, and the coating surfaces were slightly rough to the touch (Figure 3a).

At Rodão, after ten years of exposure, the anodized aluminum specimens' upward surfaces became yellow with dark-grey spots and were extremely worn and rough. However, the back surfaces retained practically the original aspects (Figure 3b).

In relation to the type of degradations processes that occurred, it was observed that the specimens exposed at Barreiro showed only pitting corrosion (Figure 4a,c), typical of the kind of atmosphere to which they were exposed—that of an industrial site with marine influence (Table 1). Pitting corrosion was more intense and extensive on the back faces, which were also dirtier. Some superficial attack of the anodic coatings might have occurred as well. It should be noticed that the bare aluminum specimens exposed at this site suffered generalized pitting corrosion (Figure 4e).

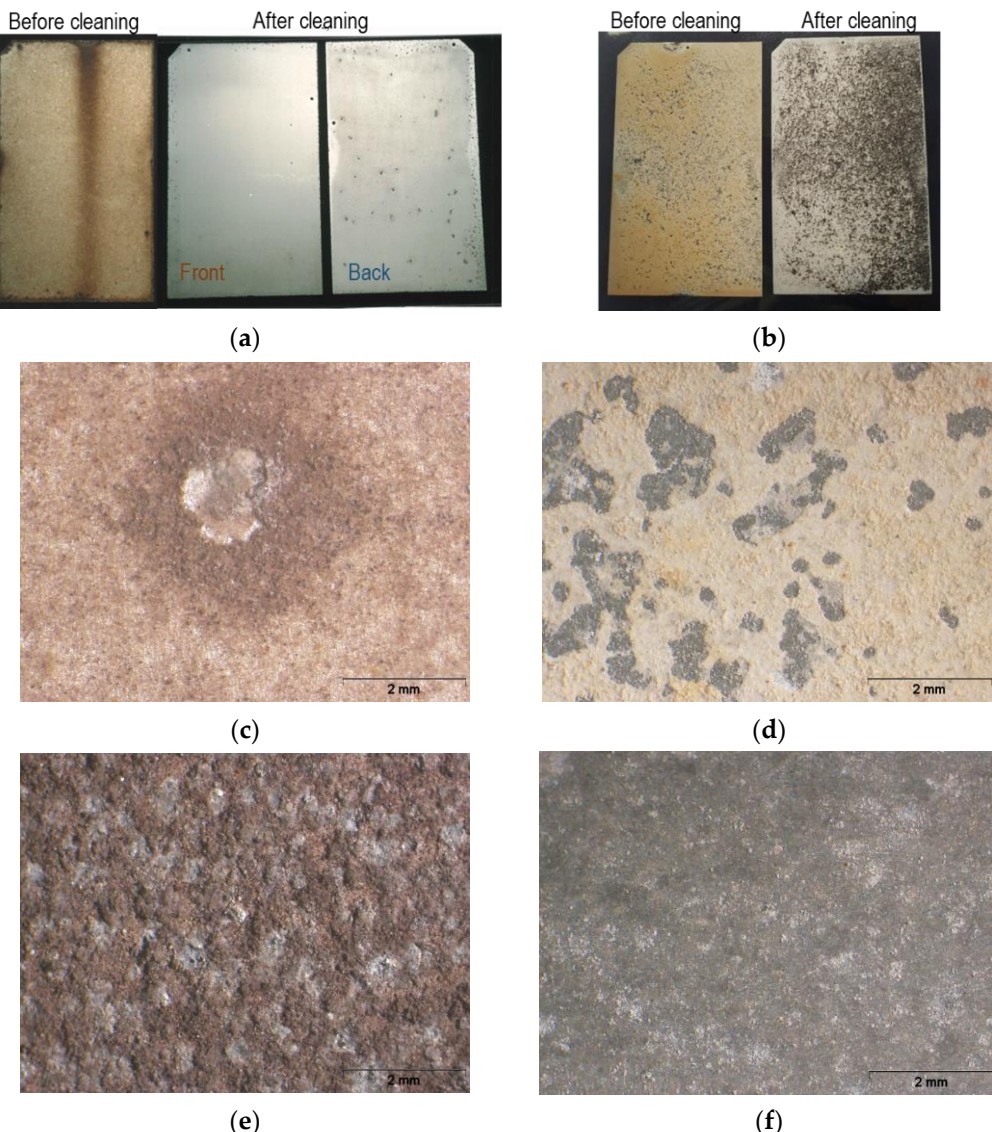

**Figure 4.** Test specimens exposed at Barreiro (on the left) and at Rodão (on the right): (**a**,**b**) visual aspects before and after cleaning (*Steps I* and *II*); (**c**,**d**) OM images of the anodized aluminum front surfaces after 10 years' exposure and (**e**,**f**) OM images of the bare aluminum specimens' front surfaces after the same time of exposure.

The anodized aluminum specimens exposed at the Rodão test site showed generalized dissolution of the coatings on the front surfaces and possible uniform corrosion of the aluminum substrates at the black spots (Figure 4b,d), which were of the same kind as those observed on the front surfaces of the bare aluminum specimens exposed (Figure 4f).

Anodic coating thickness influenced the length of time required for the test specimens to show pitting corrosion. For the lowest range (15 μm–18 μm), this took three years, while the specimens with anodic coatings with thicknesses above 30 μm, after ten years' exposure, only showed pitting at the edges of the front faces, although pitting had already occurred on the back faces. Pitting corrosion occurring at the edges of the plates was associated with small cracks that developed in these zones, probably due to the cutting that was performed to obtain the plates; therefore, they should be discounted. The maximum pit depth measured was 337 μm.

An evaluation of the anodic coatings' protective performances was carried out based on images of the type specified in ISO 10289 [36], which represent the percentages of surface areas affected by corrosion or that are damaged (A). The resultant protection ratings (Rp)

of the coatings of the specimens exposed at both test sites were plotted in the chart shown in Figure 5a. The correspondent image charts used for the classifications are included in Figure 5b. An "Rp" of 10 means that no corrosion pits or any relevant type of coating damage was observed on the surface under evaluation.

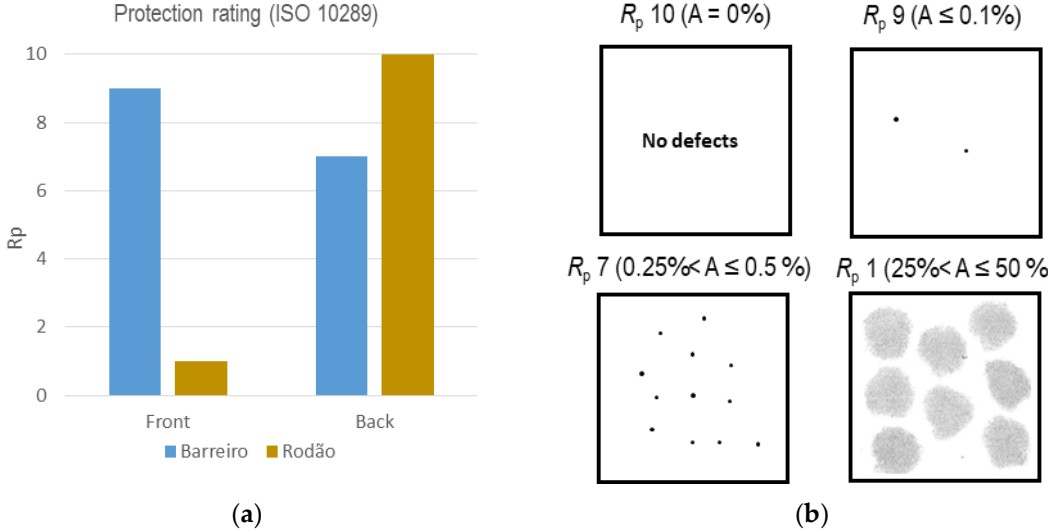

(**a**)    (**b**)

**Figure 5.** Evaluation of anodic coating protection performances after 10 years' exposure at both test sites: (**a**) average (considering all thickness ranges) protection ratings, $R_p$; (**b**) correspondent standard corrosion-affected areas and dot charts used (scale: 1:2).

*3.2. Mass Variations*

Figure 6 depicts the mass changes experienced by the test specimens exposed at the two sites plotted as a function of the time of exposure. The results obtained in a highly corrosive marine site in the same type of study [24] are included for comparison. This test site, located at Cabo da Roca, has a very wet marine atmosphere (10-year average chloride deposition rate: 194 mg·m$^{-2}$·d$^{-1}$; TOW: 5028 h·y$^{-1}$), and its corrosivity towards aluminum based on the 1st-year corrosion rate (5.1 g·m$^{-2}$) was of the C5 category.

The data used to build the graphs shown in Figure 6 were obtained by weighing the test specimens after exposure, before and after the two cleaning procedures (*Step I* and *Step II*) described above. The results of the last weighing (Figure 6c) are presented as mass losses. The following equations were used:

$$\Delta m_{exp} = \frac{m_{exp} - m_i}{A} \tag{1}$$

$$\Delta m_{wash(StepI)} = \frac{m_{StepI} - m_i}{A} \tag{2}$$

$$\Delta m_{clean(StepII)} = \frac{m_i - (m_{StepII,n} + \Delta m_{ref,n})}{A} \tag{3}$$

where $m_i$ is the original mass of the test specimens (g), $m_{exp}$ is the mass after exposure (g) before cleaning, $m_{StepX}$ is the mass obtained after the respective cleaning procedure with $n$ cycles (for *Step II*) (g) and $\Delta m_{ref,n}$ is the mass loss yielded by non-exposed anodized specimens after the same number $n$ of cycles in the *Step II* cleaning procedure (g). $A$ is the exposed area of the test specimens (m$^2$). It should be noticed that $m_{exp}$ does not include the weight of loose deposits found on the surfaces of the test specimens, since they fell away during the dismounting and transportation operations. However, at Rodão, the presence of this type of deposit on the specimen surfaces was significant during the first year of exposure. The residues accumulated (mainly wood chips) on the top of the anodized test

specimens collected after 6 months' exposure, weighed separately, ranged between 300 mg and 70 mg. This motivated the relocation of the exposure racks at one year of exposure.

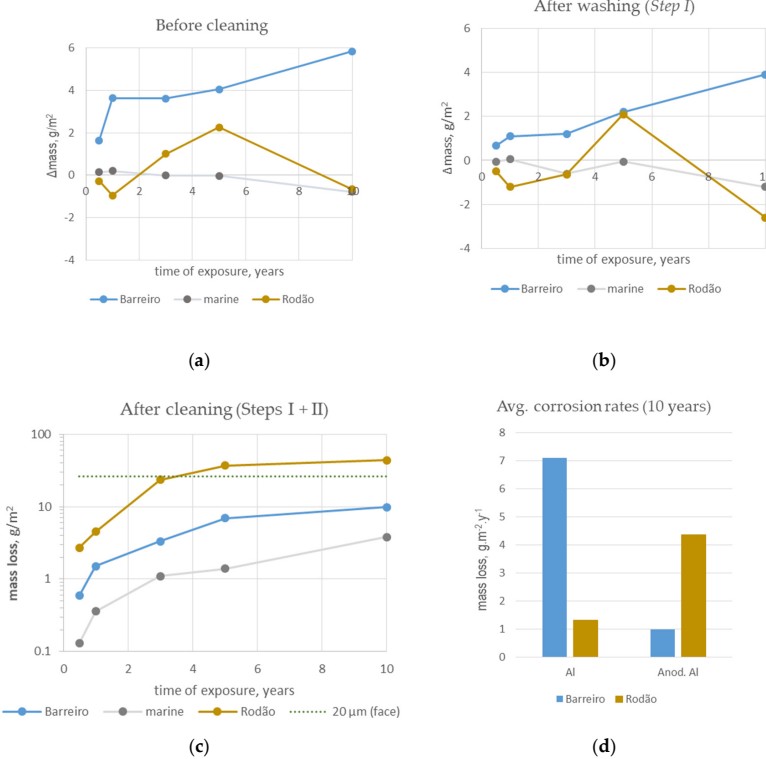

**Figure 6.** Mass variations with exposure: mass changes after (**a**) exposure, before cleaning and after (**b**) washing (*Step I* cleaning procedure); (**c**) mass loss after washing and cleaning in nitric acid (*Steps I + II* cleaning procedure); (**d**) comparison of anodized aluminum and bare aluminum corrosion rates calculated based on the mass losses obtained after ten years of exposure.

Note that the mass loss values plotted in Figure 6c are in the logarithmic form. A dotted line representing the mass loss associated with the dissolution of a 20 μm-thick anodic coating on one face was added to this graph to better perceive the degradation level.

Based on the mass losses obtained for the anodized specimens exposed at the Barreiro and Rodão sites, a "corrosion rate" for anodized aluminum was calculated for comparison with that of the bare aluminum specimens exposed at the same test sites. The results obtained for ten years' exposure are presented in the chart in Figure 6d.

### 3.3. Admittance Measurements

The measurement of the admittance of the anodic layer was carried out on one of the 10-year-exposed specimens from both test sites and on an unexposed specimen (for comparison). The results obtained are presented in Figure 7. Both the anodic coating admittance (Y) and thickness values measured were plotted instead of only the corrected admittance value for a standard 20 μm coating. This is because the anodic coating thickness measurements obtained for the Rodão test specimens' upward (front) surfaces were not reliable. The tested areas were previously washed with water to remove loose deposits, and this test was carried out in areas without corrosion pits for the test specimen from Barreiro. The admittance measurements were taken at a controlled room temperature (21 °C). It should be noted that the admittance of proper hydrothermally sealed anodic coatings should not exceed 20 μS or 25 μS [34] relative to the conventional coating thickness of 20 μm for clear (uncolored) coatings, which was the case for the samples in this study.

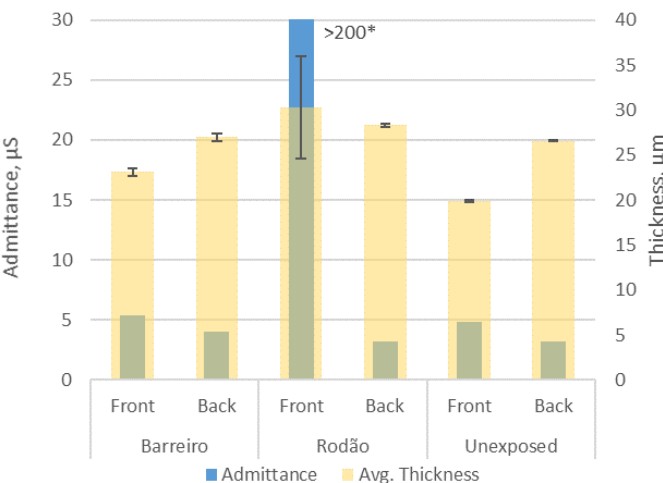

**Figure 7.** Admittances and thicknesses of the anodic coatings of the Barreiro and Rodão test specimens after 10 years of exposure and of an unexposed test specimen. The error bars plotted represent the standard deviations for anodic coating thickness measurements. * This result exceeds the measuring limit of the apparatus used.

Except for the anodic coating on the upward surface exposed at Rodão, the admittances of the anodic coatings of the exposed and unexposed specimens were similar (Figure 7) and low, ranging from 3 to 6 μS. These values are to be expected for aged coatings of good integrity [35] and implied that, besides the corrosion pits, the damage caused to the anodic coatings of the specimens exposed at the Barreiro test site was superficial and would not have affected integrity. On the other hand, the high admittance value (>200 μS) obtained for the upward face of the Rodão specimen confirmed that the anodic coating on this surface would be damaged in depth and far more degraded than those of all the others, namely, the ones on their back faces. These aspects were clarified by the following SEM observations.

### 3.4. SEM/EDS Analysis and Observations

To better understand the degradation/corrosion of the anodized aluminum specimens exposed at the two industrial test sites, SEM observations and EDS analyses were carried out for the surfaces of the exposed specimens, without cleaning, to determine the compositions of deposits and corrosion products. Additionally, cross-sectional observations were also made of these specimens to assess the anodic coating damage morphologies.

Figures 8 and 9 show the results of the SEM/EDS observations and analyses carried out on the anodized aluminum test specimens exposed for ten years at the Barreiro and Rodão test sites. SEM images of the anodic coating surfaces and cross sections from the corroded and non-corroded zones of the test specimens are presented along with the results of the SEM/EDS analyses carried out at the signaled observed zones. Optical micrographs of the same zones are also presented for better visualization of the different zones of the coating surfaces that were analyzed.

The SEM images of the anodic coating cross sections indicate the extent of anodic coating degradation. At the Barreiro test site, the coating was punctually disrupted at the corrosion pits, but outside these zones was only superficially damaged (Figure 7b). At Rodão, these observations (Figure 9b) evidenced the high degradation that occurred in the anodic coatings of the upper surfaces of the test specimens, sometimes affecting all coating thicknesses, though the anodic coatings on the back faces were still in very good condition. These findings are in agreement with the results of the admittance test.

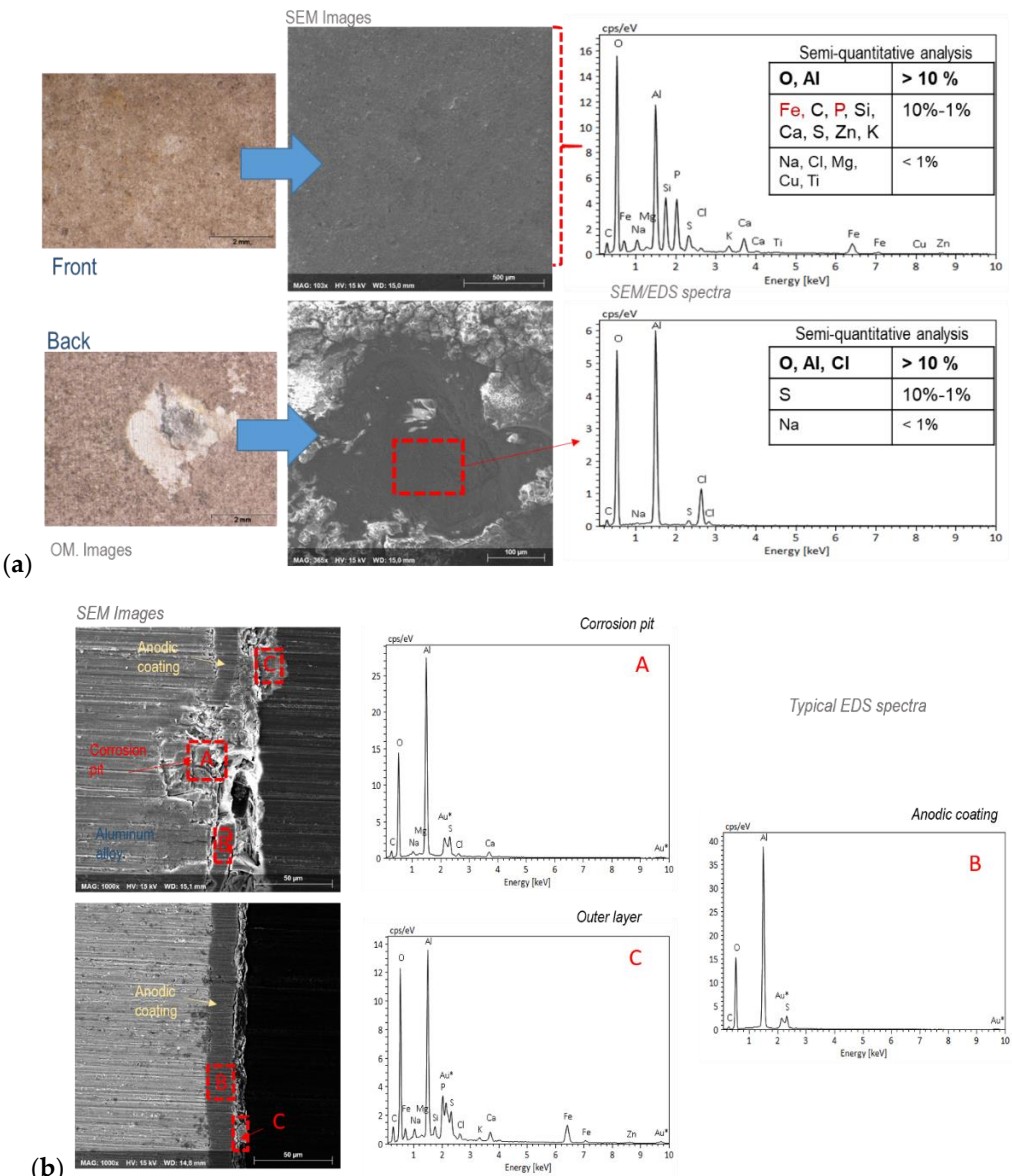

**Figure 8.** (**a**) Surface and (**b**) cross-sectional morphologies and analyses of the anodized aluminum test specimens exposed at the Barreiro test site after 10 years of exposure. The OM and SEM images were obtained from zones with and without corrosion (pitting) of the aluminum substrate; the EDS spectra and the respective semi-quantitative analysis results shown are relative to the signaled zones in the SEM images.

The EDS analyses carried out on the surfaces of the anodized test specimens exposed at Barreiro (Figure 8) revealed that, besides the elements oxygen, aluminum and sulfur, which occur naturally in the coatings, there was a significant presence of chloride (inside the pits). Outside the pits and, in general, all over the Barreiro specimens' surfaces (both faces), chloride was present in reduced amounts, and the most significant elements detected were iron, carbon, phosphorus and silicon. Calcium and sulfur could be found as minor contents, along with several other vestigial elements.

On the exposed (front) surfaces of the anodized specimens exposed at Rodão, the EDS analyses carried out (Figure 9a) revealed significant amounts of calcium and carbon, along with aluminum and oxygen. Calcium content was more relevant in the external "yellow" areas (labeled "A" in Figure 9a) of the damaged anodic coatings, while carbon was more significant in the internal "black" spots (labeled "B" in Figure 9). The other elements

detected in significant amounts in these EDS analyses were silicon and iron, along with several other vestigial elements.

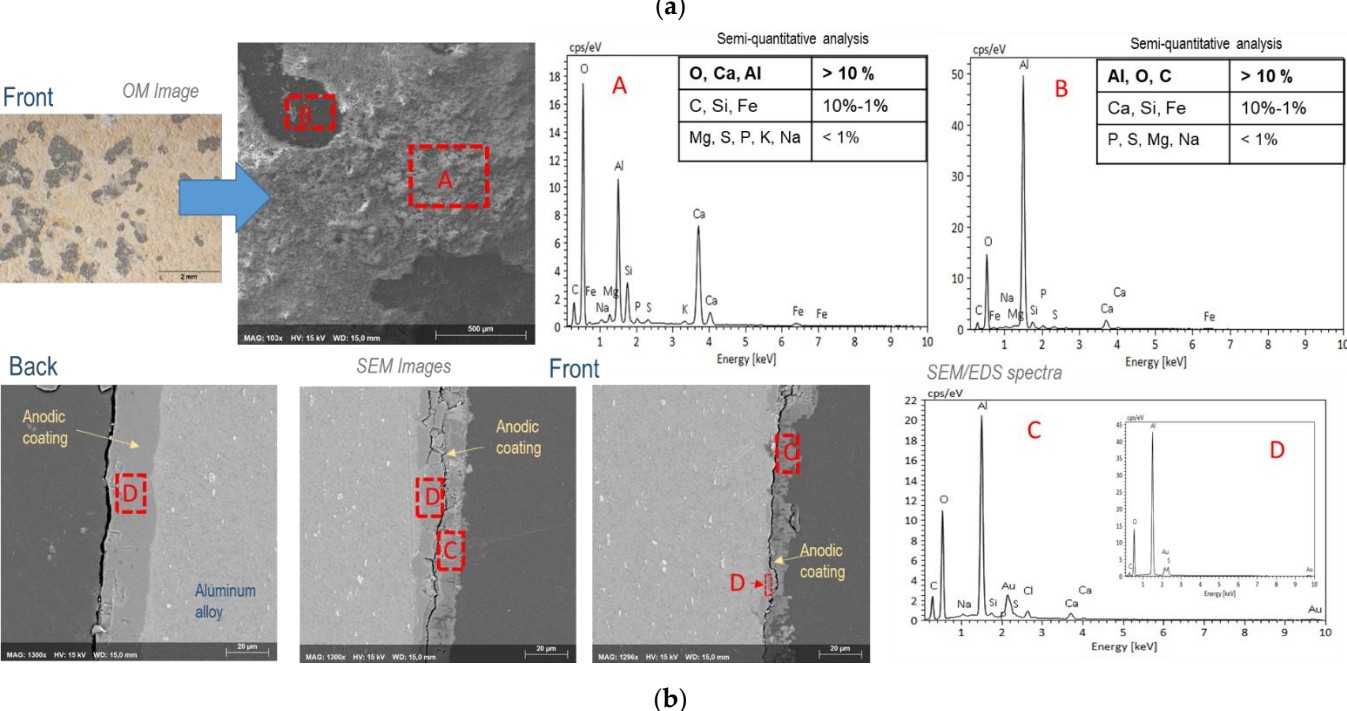

**Figure 9.** (**a**) Surface and (**b**) cross-sectional morphologies and analyses of the anodized aluminum test specimens exposed at the Rodão test site after 10 years' exposure. The OM and SEM images were obtained in zones with (front) and without (back) visible degradation of the anodic coatings; the EDS spectra and respective semi-quantitative analysis results shown are relative to the signaled zones in the SEM images.

The EDS analyses carried out in the degraded zones of the coatings (labeled "D" in Figure 9b) revealed the presence of the following elements: calcium, carbon, chloride and sodium, in addition to the elemental constituents of the anodized coatings—aluminum, oxygen and sulfur—the only ones present in the intact zones of the coatings (labeled "C" in Figure 9).

## 4. Discussion

### 4.1. Anodic Coating Aspect and Corrosion Performance

The observations made (Figures 3 and 4) revealed that the surfaces of the anodized aluminum specimens exposed for ten years at the two industrial sites presented high levels of soiling and had undergone significant visual changes and degradation processes. At Barreiro, the specimens' surfaces acquired a reddish-grey hue, showed deep pits mostly on the edges and back faces, and the coating surfaces were slightly rough to the touch. At Rodão, the skyward surfaces of the test specimens became yellow with dark grey spots and were extremely worn and rough; however, the backward surfaces were practically unaffected. The degradation processes that occurred in the anodic coatings at the Barreiro test site were mainly due to pitting corrosion, while at Rodão the coatings suffered a generalized attack. The intensities and extents of the degradation processes were higher than expected, considering only the environmental parameters (Table 1), especially at Rodão, and were therefore clearly related to the soiling levels of the surfaces and the types of deposited products.

The presence of deposits attached to the surfaces of the specimens from both test sites was indicated by the mass gains measured after exposure (Figure 6a), which were

greater for the Barreiro specimens. Weighing after washing with water and soap (Figure 6b) removed part of the deposits on the Barreiro specimens, indicating that they were loose, unlike those deposited on the Rodão specimens, which suggested that they might be integrated in the coatings. Further cleaning of the specimens with nitric acid solution (*Step II*) removed all the deposits and possibly the degraded parts of the anodic coatings, since the resultant mass losses overcame the mass gains associated with foreign product deposition (Figure 6c). The results obtained after this cleaning procedure reflected the attack suffered by the anodic coatings at both test sites and clearly evidenced the higher level of degradation of the anodized specimens exposed at Rodão, in line with what was observed visually. The average mass loss measured for the Rodão test specimens after three years of exposure was equivalent to the loss of an anodic coating more than 20 μm thick on the exposed face.

When compared with the pure marine test site, the mass variations suffered by the anodized specimens exposed at the two industrial test sites presented here were much more significant and indicated a higher level of anodic coating degradation.

When compared with bare aluminum (Figure 6d), the average corrosion rate of the anodized aluminum at the Barreiro test site was much lower, revealing the ability of the anodic oxide coating to effectively protect the substrate (Figure 5) and extend its durability, even in this very corrosive atmosphere. Although the aesthetic value was affected, the damage inflicted by pitting corrosion on the anodized aluminum would not affect the functional and technical characteristics of the aluminum alloy if it were part of a building component.

The anodic coating thickness was relevant to delaying the initiation of pitting corrosion, in accordance with what has been observed in other, similar studies: only the anodic coatings with thicknesses above 30 μm did not show pitting after ten years of exposure, but this solely obtained for the skyward faces that were subjected to the cleaning action of the rain. On the backward faces, where the accumulation of dirt on the coating surface was higher, the thicker anodic coatings did not prevent the formation of large, deep pits.

With due reservations, comparisons of the mass-loss-based corrosion rate data obtained for the anodized aluminum specimens in this study (Figure 6c,d) with similar data from other studies can be made. The yearly corrosion rate of the specimens exposed at Barreiro was 1 $g \cdot m^{-2} \cdot y^{-1}$ (~0.4 $\mu m \cdot y^{-1}$), which is within the range obtained in marine polluted atmospheres with much higher $Cl^-$ but lower $SO_2$ deposition rates [2]. In heavily polluted industrial atmospheres, values in the order of 0.4–0.9 μm/y have been found after 8 to 10 years of exposure [5,14]. The average corrosion rate of the specimens exposed at Rodão was 4.4 $g \cdot m^{-2} \cdot y^{-1}$ (~1.7 $\mu m \cdot y^{-1}$), which is higher than what is usually reported for anodized aluminum, being closer to what was observed in one case subjected to heavy soiling reported by González et al. [2].

At Rodão, the anodized aluminum specimens performed worse than bare aluminum, showing higher mass losses (Figure 6d) and increased aspect modifications (Figure 4). This unexpected behavior, especially considering the relatively low corrosion rate evidenced by the aluminum specimens, points to the contribution of unusual factors, most likely related to the nature of the products deposited on the coatings' surfaces, to the corrosion/degradation processes that occurred at this test site. The reason why the deposited products seemed to affect the anodic coating more than the aluminum alloy substrate is addressed in the following section.

*4.2. The Nature of the Deposited Products and Their Influence on the Anodic Coating Degradation Processes*

The compositions of the specimens' surface products determined by the EDS analyses that were carried out reflect the chemistries of the surrounding environments and of the coatings themselves. All the foreign elements can be related to the deposition of airborne particles of industrial waste, dust, sand and products from the corrosion of the other metallic materials exposed in the same rack.

Sulfuric anodization oxide coatings are naturally mainly constituted by aluminum and oxygen (with a mass ratio close to 1:1) and some sulfur (~5%), which makes it difficult to distinguish any products that are the result of aluminum corrosion or coating degradation from the coating products. Therefore, what will be considered for the evaluation of the degradation phenomena and their main agents are the variations in the Al:O ratios and the presence of foreign elements on the coating surfaces. For instance, the significant chloride contents found inside the corrosion pits formed in the Barreiro test specimens (Figure 7) indicate the presence of aluminum chlorides at the bottom of the pits, explaining their formation [9,20]. The presence of sulfur inside these pits may derive from anodic coating residues, but also may indicate the formation of aluminum hydroxysulfates. These are common corrosion products of aluminum in industrial atmospheres that have low solubility, not being easily leached by the action of rainwater, and hence can be expected to be found in the interiors of the highly open pits typically formed in industrial–marine atmospheres [20]. Surrounding the pits and generalized all over the surface of the Barreiro test specimens, iron, phosphorus and carbon are the most predominant foreign elements. The first two derive from iron oxides and phosphate products associated with the production of sulfuric acid (which involves the roasting of pyrite) and the production of fertilizers [37]. X-ray diffraction analyses carried out on the products present on the surfaces of the other metallic material specimens exposed along with anodized ones [23] confirmed the presence of iron compounds ($Fe_2O_3$, the main constituent of pyrite ashes and pyrite, $FeS_2$), as well as metallic phosphates and silica ($SiO_2$). The deposition of carbonaceous products, such as soot and ashes, is a possible explanation of the carbon contents of the deposits.

Pyrite ash combined with high levels of sulfur dioxide deposition can lead to the formation of extremely acidic solutions on metal surfaces [37]. It would have degraded the anodic coatings and facilitated chloride attack, leading to more intense pitting corrosion on the back surfaces, where the deposition of airborne particles was heavier. Additionally, the presence of soot and other carbonaceous particles, which are considered highly electropositive [2], must have enhanced the corrosiveness of the soiling deposits formed at the Barreiro test site and consequently contributed to its corrosivity with respect to aluminum.

The significant presence of calcium in the surface deposits found on the anodic coatings of the test specimens exposed at Rodão can also be related to industrial waste. It can be expected to result from the deposition of calcium carbonate, the main constituent of the lime sludge produced at the pulp mill—a waste product that may contain other inorganic compounds (including chlorides) and metallic elements [38]. The presence of calcium carbonate on the specimens' surfaces does not exclude the possibility that other calcium compounds might also have been deposited, namely, calcium oxide (also a product used in the pulping process), which is much more aggressive with respect to the coatings than calcium carbonate.

The significant carbon contents found in the more recessed zones of the damaged coatings of these test specimens imply the retention of other carbonaceous products in addition to lime. One possible source would be residues of wood chips pilled nearby that had been deposited on the specimens' skyward surfaces, mainly during the first year of exposure (Figure 2), before the relocation of the exposure rack.

SEM observations (Figure 9) of coating cross sections evidenced the depth of the attack of the anodic coatings that occurred on the upper surfaces of the test specimens exposed at Rodão and its relation to the above-mentioned deposited articles. The back surfaces were practically in new condition. This acute difference between the weathering of both anodic coating surfaces of the test specimens clearly demonstrates the detrimental role played by the deposited particles on the anodic coating degradation process. The type of damage inflicted on the anodic coating can only be explained by contact with highly aggressive solutions, most possibly of alkaline nature, such as those that result from calcium product dissolution when these surfaces become wet.

The presence of large amounts of wood-chip deposits over the anodized specimens exposed at Rodão during the first year of exposure should have contributed to the begin-

ning of the anodic coating degradation that occurred in this these specimens, since during wetting periods, wood (in this case, mainly *eucalyptus*) can produce acidic solutions [39,40]. These acidic solutions' attacks might have roughened the usual smooth anodic coating surfaces and hence facilitated the retention and accumulation of lime particles and products alike. These products would have then reacted with the aluminum oxide coatings, increasing even more the surface roughness and leading to greater accumulation. It was observed that the anodized aluminum test specimens retained these deposits for much longer times than the bare aluminum specimens. This may be the reason for the higher mass losses observed in the anodized specimens in comparison to the bare aluminum ones, since the residence times and numbers of aggressive particles in contact with the surfaces were, respectively, longer and higher. This test site has a relative dry climate (low TOW values Table 1), with the rainy days concentrated in one part of the year. Thus, the beneficial washing effect of rain is reduced at this site.

## 5. Conclusions

In this study, architectural anodized aluminum specimens with different coating thickness ranges were exposed at two industrial sites for up to ten years. During exposure, the test specimens were subjected to heavy deposition of different kinds of particulate matter. Based on the observations and analyses carried out, it can be concluded that the deposition of these particles, mostly of products from the industrial processes nearby, contributed largely to the degradation/corrosion of the anodized aluminum. Corrosion was more extensive than expected solely based on the usual corrosive environmental parameters and was much more significant on the surfaces with greater accumulations of deposits.

At the Barreiro test site (a conglomerate of several chemical industries, including sulfuric acid and fertilizer plants), characterized by a severe industrial atmosphere with marine influence, the deposited products were mainly constituted by iron oxides, phosphate and carbonaceous compounds. These products, combined with the high levels of sulfur dioxide deposition, acidified the surface water films formed on the specimens' surfaces and promoted the development of corrosion cells. This process enhanced the corrosive action of the chlorides present in the atmosphere, leading to the development of significant pitting corrosion, even on the thicker anodic coatings (>30 μm) under sheltered exposure conditions. Anodic coating thickness was only relevant to retarding the initiation of pitting corrosion, as found in other studies. The tests and observations carried out showed that, besides the rather localized pitting, the damage suffered by the anodic coatings in this industrial environment was superficial, being mainly of an aesthetic nature.

At the Rodão test site (that of a pulp and paper mill), considered a moderate industrial atmosphere, based on the standard aluminum corrosion rate (Table 1), the poor performance of the anodic coating was unexpected. The deposited products were mainly constituted by calcium compounds and wood particles. Some calcium compounds when wetted produce strong alkaline solutions that are known to be highly detrimental to the anodic coating, promoting its rapid dissolution. It was proposed that the presence of wood-chip deposits in the first year of exposure at this site, which can yield acidic solutions during wetting periods, may have additionally contributed to the intense anodic coating degradation that occurred in the following years. The level of damage presented by the anodic coatings in this specific industrial environment was independent of thickness and was far more than aesthetic. The kind of degradation that occurred would possibly impair the functionality of some building components, namely, sliding windows frames made of this material.

The results obtained in this atmospheric corrosion study confirmed that anodization can be very effective in protecting aluminum against corrosion in severely polluted industrial sites, even with high soiling effects, presenting only some aesthetic damage. The level of protection conferred will increase with anodic coating thickness and cleaning frequency (e.g., exposure to rain-washing effects). However, in the presence of alkaline deposits, the protective ability of the anodic coating is strongly compromised. This type

of effect has also been found in anodized aluminum building components, when careless installation procedures have put them in contact with cementitious materials in new constructions [3,41].

The work presented here additionally aims to draw the attention of architects, designers and other users to the need to consider the risk of soiling deposition in the assessment of the durability of anodic coatings, as well as to the importance of implementing cleaning maintenance works for the assurance of their durability.

**Author Contributions:** Investigation, methodology, data treatment, formal analysis, writing—original draft preparation, I.R.F.; writing—review and editing, E.E. All authors have read and agreed to the published version of the manuscript.

**Funding:** This research received no external funding.

**Data Availability Statement:** Not applicable.

**Acknowledgments:** The authors wish to thank the experimental support given by Ana Paula Menezes and Ana Paula Melo, technicians of the Metallic Materials Division of LNEC, in the preparation of test specimens and SEM/EDS observations.

**Conflicts of Interest:** The authors declare no conflict of interest.

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
