# Peer review of "Influence of Exposure Conditions and Particulate Deposition on Anodized Aluminum Corrosion"

_cmd, doi:10.3390/cmd3040040_

Round 1

Reviewer 1 Report

Authors report technical purity aluminum protected by sulfuric acid anodizing. what is very exciting about the paper, they show exploitation of the coatings in real environment - in two locations in Portugal, what is much more interesting for industry than standard NSST tests.

Nevertheless, I would like to ask Authors would it be possible to supplement the manuscript with:

- polarization tests of coatings and discuss obtained corrosion potential, corrosion current density and pitting potential with the observations described in the manuscript

- perform EIS screening like |Z| at 0.01 Hz and also discuss with the results.

After this mdoerate revision, the papers is suitable for publishing.

Reviewer 2 Report

Fig. 6a-c, marine atmosphere included in the graph was not described before. Please provide details of the exposure site.

Please specify what anodic film thickness the panels in Fig.6 had. If the data correspond to a weighted result from three differnt thicknesses, please specify how it was calculated. 

Reviewer 3 Report

1.    All of photos

You should show scales on these photos. 

2.    Table 2

Is there any difference in rainfall between these two areas for exposure tests? This will be related to cleaning action of the rain. You should discuss about this. 

3.    Caption of Fig.7 

You must describe what the three images at the bottom show.

4.    Caption of Fig.8 

You must describe what the four images at the bottom show.

5.    L. 396 “At Rodão, the anodized aluminum specimens…. processes occurred in this test site.”

I can’t understand why “anodized aluminum specimens performed worse than bare aluminum, showing higher mass losses”? You should discuss how contribute nature of the products deposited on corrosion protection of bare Al and anodized Al?

6.    L. 483 “L. 396 "At Rodão test site…. degradation observed."

I guess this will affect not only to anodic oxide film, but also to bare Al. Please discuss about this. 

Reviewer 4 Report

This study presents the atmospheric corrosion of the anodized aluminum exposed in two different chemical industrial complexes.The data was observed during a 10 years’ atmospheric corrosion study.

The paper lacks some data to prove the theoretical discussion, such as the acid-base property of hydrolysis, but it is significant. The data accumulated in the past ten years is worth publishing.

Figure 5 shows the resultant protection rating (Rp) of the coating obtained for the specimens exposed at both test sites, Please add an explanation about Rp

Round 2

Reviewer 1 Report

I would like to recommend at least polarization test of as-prepared samples.

Reviewer 3 Report

I think this paper is interesting and useful for many fields. 
